# Relating a System’s Hamiltonian to Its Entropy Production Using a Complex Time Approach

**DOI:** 10.3390/e25040629

**Published:** 2023-04-06

**Authors:** Michael C. Parker, Chris Jeynes

**Affiliations:** 1School of Computer Sciences & Electronic Engineering, University of Essex, Colchester CO4 3SQ, UK; mcpark@essex.ac.uk; 2Ion Beam Centre, University of Surrey, Guildford GU2 7XH, UK

**Keywords:** QGT, entropic Hamiltonian, Bekenstein–Hawking relation, analytical continuation, Loschmidt Paradox, Riemannian geometry, Kramers–Kronig relations, arrow of time

## Abstract

We exploit the properties of complex time to obtain an analytical relationship based on considerations of causality between the two Noether-conserved quantities of a system: its Hamiltonian and its entropy production. In natural units, when complexified, the one is simply the Wick-rotated complex conjugate of the other. A Hilbert transform relation is constructed in the formalism of quantitative geometrical thermodynamics, which enables system irreversibility to be handled analytically within a framework that unifies both the microscopic and macroscopic scales, and which also unifies the treatment of both reversibility and irreversibility as complementary parts of a single physical description. In particular, the thermodynamics of two unitary entities are considered: the alpha particle, which is absolutely stable (that is, trivially reversible with zero entropy production), and a black hole whose unconditional irreversibility is characterized by a non-zero entropy production, for which we show an alternate derivation, confirming our previous one. The thermodynamics of a canonical decaying harmonic oscillator are also considered. In this treatment, the complexification of time also enables a meaningful physical interpretation of both “imaginary time” and “imaginary energy”.

## 1. Introduction

The physical interpretation of time, and the peculiar characteristics of the arrow of time, have long excited scientific curiosity and generated perplexing puzzles. Alexander Klimenko [1] recently summarized the issues (with extensive references) in the context of a deep discussion of quantum decoherence as an indicator of irreversibility. We will approach the issues from a rather different point of view, accepting the standard treatments of *causality*.

The Second Law of Thermodynamics has a problematical status: is it fundamental or emergent? Is irreversibility an emergent property determined at the microscopic level? (We could also ask, again related to the intrinsic nature of time, is the universe essentially non-local?) Currently, the fundamental equations of quantum mechanics (QM) are time reversibility, and the elusiveness of an *emergent* irreversibility (generating the Second Law) is known as the Loschmidt Paradox. On the other hand, most physicists assert that the Second Law is fundamental. How is this state of affairs to be squared? It transcends the other well-known problem: the inherent incompatibility between the realms of QM (the physics of the very small) and general relativity (GR, the physics of the very large). Here, we show how complexifying time may help resolve both of these problems.

We should remark that the Loschmidt Paradox has recently been resolved by Tessarotto et al. [2] for the case of (dilute) ideal gases. Their result is actually remarkable because they prove (under certain conditions) that Boltzmann’s *H*-theorem is *time-reversible*, and thus that the resulting increase in entropy is also time-reversible. However, there remain a number of caveats for this result, the main one being that the continuous limit appears to be an intrinsic requirement but introduces unresolvable inconsistency. It should also be mentioned that ideal gases are not the only systems of interest. Our treatment here is of the general case.

Imaginary time is invoked in discussions of quantum mechanical tunnelling times (see, for example, McGlynn and Simenel 2020 [3]), although its physical interpretation remains obscure. However, it is surprising that the concept of a complex temporal *plane* has not been suggested until very recently [4], perhaps because it seems intuitive that time is essentially uni-dimensional, in contrast to the obvious multi-dimensionality of space with its rich geometry. It should be noted that the complexification of time is well-known; that is, Minkowksi found that a complex description for spacetime elegantly satisfies the geometrical requirements of special relativity; he defined time as imaginary, in contrast to real space (with unit vectors i*t* ≡ i*x*_0_/*c*, *x*_1_, *x*_2_, *x*_3_, where, as usual, *c* is the speed of light and i ≡ √ − 1, such that the squares of the unit vectors are −1, +1, +1, and +1, respectively, or simply “−+++”). However, the *inverse* metric for spacetime (real time and imaginary space: “+−−−”) is also valid, and we will show where each metric is physically appropriate.

We also note the clear isomorphism between the Hamiltonian *H* of a system and its entropy production P. The Hamiltonian of a system describes its energy content and, on the assumption of a lossless (reversible) physical process, the First Law of Thermodynamics trivially applies and the Hamiltonian is conserved. This is in accordance with Noether’s theorem based on the Euler–Lagrange equation for the Principle of Least Action, exploiting *time* as the key symmetry. In the same way, we have previously shown [5] that entropy production must also be a conserved quantity by Noether’s theorem, this time applied to the *entropic* Euler–Lagrange equation associated with the isomorphic Principle of Least Exertion [6] (again using the time symmetry). Thus, it is clear that the Hamiltonian and entropy production are isomorphic to each other, both being Noether-conserved quantities. We will show here that they are also directly related, physically. 

The paper will treat alpha particles and black holes, because, in quantitative geometrical thermodynamics (**QGT**; Parker & Jeynes 2019 [6]), they have a similar description: both have an entropy determined by the Bekenstein–Hawking relation, a relation that is a necessary (holographic) consequence of the *entropic* Liouville theorem (see Parker and Jeynes 2021a [7]). Both are *unitary entities* in QGT, that is, there exist no simpler entities at that scale. Curiously, although both are maximum entropy entities (being unitary), the one has zero entropy production and the other has positive (non-zero) entropy production. However, even though the black hole necessarily grows, it still remains the same unitary entity. The entropy production of black holes was calculated previously (Parker and Jeynes 2021b [5]); here, we present an alternate derivation of the same result. It turns out that the matter radius of the alpha (and other nuclei) is readily calculated by QGT (Parker et al. 2022 [8]), thus it is helpful to show that these complex time methods are also valid for what might have been thought a trivial case.

## 2. Complex Time Descriptions for Reversibility and Irreversibility

The fundamental kinematic equations,
(1a)$$\frac{\partial }{\partial t}\Phi =2k_{B}D\nabla^{2}\Phi \;\mathrm{and}\;$$
(1b)$$-i\frac{\partial }{\partial \tau }\Psi =\frac{h}{4\pi m}\nabla^{2}\Psi \;$$
are the diffusion equation (describing the irreversible evolution of the probability amplitude F, and involving Boltzmann’s constant k_B_ and the diffusion coefficient D) and Schrödinger’s equation (describing the reversible evolution of the quantum wave function Ψ, and involving Planck’s constant *h* and the mass *m* of the particle, respectively); note that both equations are temporal in character, but we employ different symbols for their respective time variables, because Equation (1a) is essentially real in nature, whereas Equation (1b) is imaginary. Although different, using the diffusion coefficient definitions of Mita [9] and Nelson [10], the above two equations are related by the following:(2a)t↔iτ
(2b)$$\mathrm{and}\;\frac{h}{4\pi m}↔\;2k_{B}D\equiv \gamma$$
suggesting that, owing to the isomorphism between Schrödinger’s equation and the diffusion equation (Equation (1)), the real time *t* is isomorphic to the imaginary time *τ*. Equivalently, in Equation (2a), the factor i (≡√−1) looks like a Wick rotation (see O’Brien’s helpful discussion [11]), while Equation (2b) reinterprets the diffusion coefficient *D* as being like an inverse mass, *D* ↔ 1/2*m* (interpreting the 4π as the appropriate solid angle of the full sphere). The factor 2 that appears in Equations (1a) and (2b) in front of the Boltzmann constant *k_B_* is a reflection of an explicitly entropic version of the partition function, being composed of probabilities that are analogous to the modulus-squared of Schrödinger’s equation (see Parker and Jeynes 2021 [7], Equation (14d)), as is also described by Córdoba et al. 2013 [12] (see their Equations (29) and (30)).

The Wick rotation was implicitly described (in terms of Hodge duals) in Appendix A of Parker and Jeynes 2019 [6], while the notation $\hat{\eta }=2k_{B}\nabla$ was employed by Velazquez [13] to provide a commutator-like derivation of the thermodynamics uncertainty relations (again, notice the presence of the factor of 2 in Velazquez’s expression for $\hat{\eta }$). This example illustrates the Wick rotation appearing to map a reversible dynamics into an irreversible one, as already emphasized by Córdoba et al. [12].

Thus, the Schrödinger equation can be considered either as a non-dissipative wave equation (reversible) or as a diffusion equation (irreversible). The key distinction between whether the Schrödinger equation describes a reversible or irreversible process lies in whether the associated diffusion coefficient γ ≡ *ħ*/2*m* (inversely related to the friction coefficient, and now using the reduced Planck constant *ħ* ≡ *h*/2*π* for convenience) is real or imaginary. In a conventional diffusion process (such as involving Brownian motion), the diffusion coefficient is a real quantity, as described by Nelson [10], for example (see also Fritsche and Haugk [14]). However, Nelson shows that, if the diffusion constant were to be imaginary, then the average diffusion velocity would become imaginary, and a Brownian motion with zero friction is described. As a phenomenological description, the diffusion coefficient is conventionally described using the reduced Planck constant. However, exploiting the isomorphism between Equations (1) and (2) (where the Planck and Boltzmann constants are related to each other via a Wick rotation, −i*ħ*⇔2*k_B_*), it is clear that an imaginary diffusion coefficient can also be described: γ = i*k_B_*/*m*. Of course, this begs the question of what determines whether the “diffusion coefficient” is real (describing an irreversible process) or imaginary (reversible).

We are thus encouraged to define physical time as a *complex dynamical variable*; that is to say, rather than considering the (imaginary) parameter *τ* simply as isomorphic to the real time variable *t*, we now define *τ* as the *analytic continuation* of real time *t* into the complex plane. As it is convenient (following Minkowski) to assume that conventional (reversible) time *t* is imaginary, we also rotate the complex time plane by a factor i:(3)z ≡i(t+iτ)=−τ+it

In this definition of complex time *z*, we have thus *analytically continued* time *t* into an overall *complex time* measure (using the time parameter τ) followed by an overall rotation by a factor i ≡ √−1. Whereas in Equation (2a), we suggested the isomorphism t↔iτ, in Equation (3), we now explicitly distinguish the real and imaginary components of complex time *z*, such that *t* and τ appear essentially independent of each other. 

To avoid confusion, we should comment, parenthetically, that whether τ is regarded as real (or imaginary) and *t* is regarded as imaginary (or real) depends on which of the two metrics, (+−−−) or (−+++), is being used. Both are valid: the application determines which is more convenient.

Equation (15) of Velazquez et al. 2022 [4] specifies an effective equivalence between the normal Hamiltonian (representing the energy of the system) and the “entropic Hamiltonian”, both of them *complex valued*. It is tempting to identify the “entropic Hamiltonian” with entropy production, but there is some subtlety because an unconditionally *stable* system (such as an alpha particle, for which the entropy production is identically zero) would appear to be a counter example, as the Hamiltonian of the alpha is non-zero.

Nevertheless, we will show in what way the (complexified) entropic Hamiltonian of the previous treatment of [4] can be identified with (complexified) entropy production, and in what way the (energy) Hamiltonian can be identified with entropy production. This demonstration depends on the properties of analytical continuation, and is thus a consequence of the systematic complexification of the formalism. It is a deeply surprising result, intimately related to the fact that these properties are built on a holomorphic representation of the maximum entropy state. Analytical continuation is a powerful technique used in conjunction with Riemannian geometry and holomorphic functions, both of which are essential to the quantitative geometrical thermodynamics formalism (**QGT**; Parker and Jeynes 2019 [6]).

We will also show that an unconditionally *unstable* system far from equilibrium (such as a black hole, for which the entropy production is non-zero and positive) is related. QGT treatments are available for both the alpha particle (Parker et al. 2022 [8]) and the black hole (Parker and Jeynes 2021 [5]); both are unitary entities (in QGT terms), being specified by only four numbers (mass, spin, charge, and a length scale factor). We will also show that the real (dissipative) harmonic oscillator is also easily represented in this formalism.

Following Velazquez et al. [4] (*q.v.* for a wider discussion of the physical motivation), we define the quantity *actio-entropy* S as a holomorphic function of the classical action *S_cl_* and the thermodynamic entropy *S_th_* across complex time *z* ≡−τ+it (Equation (3)), which explicitly combines *action* and *entropy*. It is the unifying idea of *complex time* that allows the application of complex function theory across the complex temporal plane to coherently define the (dimensionless) analytic function S (see Equation (7) of [4]):(4)$$S=\frac{S_{th}}{2k_{B}}+i\frac{S_{cl}}{\hbar }$$

The fact that the actio-entropy S is holomorphic is shown by the validity of the corresponding Cauchy–Riemann equations, which in turn allows analytical continuation between the classical action *S_cl_* and the thermodynamic entropy *S_th_* (see Equation (14) of [4]):(5a)$$\frac{1}{2k_{B}}\frac{\partial S_{th}}{\partial \tau }=-\frac{1}{\hbar }\frac{\partial S_{cl}}{\partial t}$$
(5b)$$\frac{1}{2k_{B}}\frac{\partial S_{th}}{\partial t}=\frac{1}{\hbar }\frac{\partial S_{cl}}{\partial \tau }$$

Equations (5) imply the (conventional) fundamental definitions of the system’s energy-based (kinematic) Hamiltonian *H* as a function of the classical action, as well as the entropy production Π (rate of increase of entropy), as defined using the respective reversible (*t*) and irreversible (τ) temporal axes (see Equation (4) of [4]):(6)$$H=-\frac{\partial S_{cl}}{\partial t}↔\;\Pi =\frac{\partial S_{th}}{\partial \tau }$$

Exploiting the symmetries associated with the Wirtinger operator [15], the definition of differentiating across the complex plane *z* is given by the following (see Equation (A56) of [4]):(7)$$\frac{\partial }{\partial z}\equiv \frac{1}{2}\left(\frac{\partial }{\partial \tau }+i\frac{\partial }{\partial t}\right)$$

We see that the (energy) Hamiltonian and entropy production are each associated with both real and imaginary components, according to the respective temporal domains of the complex time plane (*z* ≡ −τ + i*t*; see Equations (A57) and (A58) of [4]) and considering the complex differential of the actio-entropy ∂S/∂z:(8a)$$H_{z}=2i\frac{\partial S_{cl}}{\partial z}=-\frac{\partial S_{cl}}{\partial t}+i\frac{\partial S_{cl}}{\partial \tau }\equiv H+iH_{\tau }$$
(8b)$$\Pi_{z}=2\frac{\partial S_{th}}{\partial z}=\frac{\partial S_{th}}{\partial \tau }+i\frac{\partial S_{th}}{\partial t}\equiv \Pi +{i\Pi }_{t}$$

We emphasise that the symbol *H* represents the real (reversible) part of the complex Hamiltonian *H_z_* (where the subscript *z* indicates its relation to the complex temporal plane *z*) and Π is the real (irreversible) part of the complex entropy production Π*_z_*. Note that, where Velazquez et al. [4] use the term “entropic Hamiltonian”, we use instead the term “entropy production” as a synonym for “*rate* of entropy increase”.

The role of the (complex) Hamiltonian *H_z_* and (complex) entropy production $\Pi_{z}$ in any physical process must be understood from the trajectory across complex time taken by the physical phenomenon. That is, whether the process is reversible or irreversible (or a mixture) will be determined by how the real and imaginary components are combined at each point in time. We point out parenthetically that the representations we use seem ambiguous at this stage as to whether reversibility is indicated by the real or by the imaginary components. However, the usual convention is to regard the imaginary temporal *t*-axis as the reversible one, in the context of a complex temporal plane providing the comprehensive framework in which both reversible and irreversible processes can be consistently and completely described.

In Equation (8), the complex expressions for *H_z_* and $\Pi_{z}$ are comparable to the expressions used in signal processing for analytic quantities (in particular, photons). QGT [6] shows how meromorphic functions are used to express information and how holomorphic functions express maximum entropy systems. John Toll [16] explicitly gives a rigorous proof that strict causality is logically equivalent to the existence of the “dispersion relations”, which are best known as practical constraints in signal processing, so that, in optics (for example), the refractive index has an imaginary component in the presence of absorption. However, as any particle can be represented as a wave, any scattering process must have a representation in terms of a “frequency distribution”, with the corresponding “group” and “phase” velocities. Toll has shown how the real and imaginary properties of the dispersion are mutually related via the Kramers–Kronig relations, using the properties of the Hilbert transform. Toll further points out that exactly the same formalism is applicable generally; not only to optics but also to (for example) high-energy particle scattering (citing the “excellent discussion” of the so-called “R-matrix” representation by Wigner [17]).

In the optics example, the real dielectric component is associated with zero absorption (a thermodynamically reversible phenomenon) and the imaginary component (absorption, or indeed amplification) is associated with thermodynamic irreversibility. To be more specific, we note that Fourier theory requires the association of the complex time plane *z* ≡ −(*τ* − i*t*) = i(*t* + i*τ*) with its counterpart (complex-conjugated) complex frequency plane $\hat{\omega }=-i\left(\omega -i\upsilon \right)=-\left(\upsilon +i\omega \right)$.

## 3. Fourier and Hilbert Transform Relations

We define the conjugate frequencies of the respective real and imaginary temporal components consistently using the Fourier transform definition:(9)$$F\left(\omega \right)=\overset{\infty }{\underset{-\infty }{\int }}f\left(t\right)e^{i\omega \cdot t}dt\to F\left(\hat{\omega }\right)=\overset{\infty }{\underset{-\infty }{\int }}f\left(z\right)e^{i\hat{\omega }\cdot z}dz$$

That is, both the time *t* and frequency ω are respectively analytically continued into their appropriate complex planes, with the complex time *z* and complex frequency $\hat{\omega }$ representing the appropriate conjugate pair, $\hat{\omega }⇋z$, with the consistent Fourier transform relation given above.

For the case when the function *H*(*t*) is causal (that is, *H*(*t*) = 0 for *t* < *t*_0_, where *t*_0_ is chosen as a convenient point in time to express the causality of the system) and is a physically realisable (square-integrable) function, then Cauchy’s theorem applies, and the Hamiltonian is holomorphic in the required (upper, as appropriate) half-plane, such that it obeys the dispersion relations. Following Toll [16] Equation (2.5), we can write the dispersion of the complex Hamiltonian (using the terms of Equation (8a) above) in terms of the component *ω* of the complex frequency $\hat{\omega }$:(10a)$$H\left(\omega \right)=+\frac{1}{\pi }P\overset{\infty }{\underset{-\infty }{\int }}\frac{H_{\tau }\left(\omega^{\prime }\right)}{\omega^{\prime }-\omega }d\omega^{\prime }$$
(10b)$$H_{\tau }\left(\omega \right)=-\frac{1}{\pi }P\overset{\infty }{\underset{-\infty }{\int }}\frac{H\left(\omega^{\prime }\right)}{\omega^{\prime }-\omega }d\omega^{\prime }$$
where “P” is the principal part to be taken at the point $\omega^{\prime }$ = ω. Note that Toll explicitly emphasises that, because the real and imaginary parts of *H_z_* (see (Equation (8a)) are Hilbert transforms of each other (Equation (10)), they are indeed *causal*. The corollary is that Equation (10a) implies Equation (10b), and *vice versa*. The integration is performed parallel to the *ω*-axis and the analytical continuation into the upper half-plane exists.

The Cauchy–Riemann relations, Equation (5), written as functions on complex time, clearly imply complementary relations in terms of the complex frequency. Because frequency is essentially inverse time, there is a very close relationship between the entropy production and (energy) Hamiltonian, where the entropy production (being most closely associated with non-reversible, dissipative processes) is aligned with the irreversible (real) temporal *τ*-axis, and the Hamiltonian (being most closely associated with reversible, non-dissipative processes) is aligned with the reversible (imaginary) temporal *t*-axis. Equivalently, for the conjugate frequency axes, the entropy production is intrinsically associated with the real frequency *υ*-axis, while the Hamiltonian is intrinsically associated with the imaginary frequency *ω*-axis.

When we analytically continue the entropy production or the Hamiltonian from one axis across the complex plane to the orthogonal axis, we exploit the symmetries that manifest in the mathematics. The key measurables are the real part of the energy (associated with the reversible axis) and the real part of the entropy production (associated with the irreversible axis). From this perspective, when transforming a quantity from one axis to the other, that is, finding the Hilbert transforms of the entropy production and the (energy) Hamiltonian (such transforms can also be regarded as a kind of Wick rotation), the result is particularly useful for interpreting what might be considered as the ‘cross-axial’ terms (Π*_t_* and *H*_τ_, see Equations (8), and we shall see that Parseval’s Theorem, as applied to the respective Hilbert transform components, also provides additional useful insight into their physical properties). Thus, exploiting the mathematical properties associated with the process of analytical continuation, we may write two symmetrical pairs of expressions for how the complex entropy production function and the complex Hamiltonian function relate along the two conjugate frequency axes forming the complex frequency plane:(11a)$$\Pi_{t}\left(\omega \right)=-i\Pi \left(\upsilon \right)$$
(11b)$$\Pi \left(\omega \right)={i\Pi }_{t}\left(\upsilon \right)$$
(11c)$$H_{\tau }\left(\upsilon \right)=-iH\left(\omega \right)$$
(11d)$$H\left(\upsilon \right)=iH_{\tau }\left(\omega \right)$$

Using the Cauchy–Riemann relations of Equation (5), we can now relate the entropy production values on the real frequency *υ*-axis to the appropriate energy Hamiltonian values on the imaginary frequency *ω*-axis:(12a)$$\frac{1}{\hbar }H\left(\omega \right)=-i\frac{1}{2k_{B}}\Pi \left(\upsilon \right)$$
(12b)$$\frac{1}{\hbar }H\left(\upsilon \right)=i\frac{1}{2k_{B}}\Pi \left(\omega \right)$$
(12c)$$\frac{1}{\hbar }H_{\tau }\left(\omega \right)=i\frac{1}{2k_{B}}\Pi_{t}\left(\upsilon \right)$$
(12d)$$\frac{1}{\hbar }H_{\tau }\left(\upsilon \right)=-i\frac{1}{2k_{B}}\Pi_{t}\left(\omega \right)$$

Applying Equations (12c) and (11b) to Equation (10b) immediately shows that the entropy production component, as observed on the (reversible time) *t*-axis, is the Hilbert transform of the corresponding Hamiltonian:(13)$$\frac{1}{2k_{B}}\Pi \left(\omega \right)=-\frac{1}{\hbar }\frac{1}{\pi }P\overset{\infty }{\underset{-\infty }{\int }}\frac{H\left(\omega^{\prime }\right)}{\omega^{\prime }-\omega }d\omega^{\prime }$$

Further, using Equations (8a), (10b), and (13) allows us to express the complex Hamiltonian $H_{z}\left(\omega \right)$ along the same reversible time axis:(14)$$\frac{1}{\hbar }H_{z}\left(\omega \right)=\frac{1}{\hbar }H\left(\omega \right)+i\frac{1}{2k_{B}}\Pi \left(\omega \right)$$
(15)$$H_{z}\left(\omega \right)=H\left(\omega \right)-i\frac{1}{\pi }P\overset{\infty }{\underset{-\infty }{\int }}\frac{H\left(\omega^{\prime }\right)}{\omega^{\prime }-\omega }d\omega^{\prime }$$

That is to say, the components of the Hamiltonian and entropy production quantities along the (reversible time) *t*-axis represent the real and imaginary components, respectively, of an overall *causal* expression (with both components representing Noether-conserved quantities).

An exactly similar treatment, comparing Equations (10) to Equations (16), and Equations (13)–(15) to Equations (17)–(19), applies along the (irreversible time) *τ*-axis (although noting that its ‘causal’ properties are now in the *negative* temporal direction, as per Equation (3)), so that the signs of Equations (16) are inverse to that of Equations (10) using the associated (conjugate) real frequency component, υ, of the complex frequency plane (where “P” is the principal part to be taken at the point $\upsilon^{\prime }$= υ):(16a)$$\Pi \left(\upsilon \right)=-\frac{1}{\pi }P\overset{\infty }{\underset{-\infty }{\int }}\frac{\Pi_{t}\left(\upsilon^{\prime }\right)}{\upsilon^{\prime }-\upsilon }d\upsilon^{\prime }$$
(16b)$$\Pi_{t}\left(\upsilon \right)=+\frac{1}{\pi }P\overset{\infty }{\underset{-\infty }{\int }}\frac{\Pi \left(\upsilon^{\prime }\right)}{\upsilon^{\prime }-\upsilon }d\upsilon^{\prime }$$

It is also interesting to note that mathematically applying the Hilbert transform to the *negative* temporal direction of the irreversible *τ*-axis *inverts* the conventional cause and effect relationship that occurs in the forward time direction. That is to say, for the irreversible *τ*-axis, from the perspective of the normal (forward-propagating) direction in time, the ordering is inverted: the “effect” occurs before the “cause”. We argue that this is indistinguishable (empirically equivalent) to what is observed for apparently spontaneous or random phenomena. That is to say, our treatment of the irreversible *τ*-axis offers a description phenomenologically consistent with what is conventionally associated with entropy-increasing (irreversible) processes such as radioactivity, decoherence, spontaneous emission, and other noise-like phenomena, all of which are ubiquitous in the physical realm. The implications of this approach (that is, the relationship between what are, in effect, advanced waves and random phenomena) continue to be the topic of future investigation.

Thus, we find the following:(17)$$-\frac{1}{\hbar }H\left(\upsilon \right)=\frac{1}{2k_{B}}\frac{1}{\pi }P\overset{\infty }{\underset{-\infty }{\int }}\frac{\Pi \left(\upsilon^{\prime }\right)}{\upsilon^{\prime }-\upsilon }d\upsilon^{\prime }$$
(18)$$\frac{1}{2k_{B}}\Pi_{z}\left(\upsilon \right)=\frac{1}{2k_{B}}\Pi \left(\upsilon \right)-i\frac{1}{\hbar }H\left(\upsilon \right)$$
(19)$$\Pi_{z}\left(\upsilon \right)=\Pi \left(\upsilon \right)+i\frac{1}{\pi }P\overset{\infty }{\underset{-\infty }{\int }}\frac{\Pi \left(\upsilon^{\prime }\right)}{\upsilon^{\prime }-\upsilon }d\upsilon^{\prime }$$

From Equation (8a), we have Equations (20) (below), and from Equations (8b) and (11), we have Equations (21) (below):(20a)$$H_{z}\left(\omega \right)\equiv H\left(\omega \right)+iH_{\tau }\left(\omega \right)$$
(20b)$$H_{z}\left(\upsilon \right)\equiv H\left(\upsilon \right)+iH_{\tau }\left(\upsilon \right)$$
(21a)$$\Pi_{z}\left(\upsilon \right)\equiv \Pi \left(\upsilon \right)+{i\Pi }_{t}\left(\upsilon \right)$$
(21b)$$\Pi_{z}\left(\omega \right)\equiv \Pi \left(\omega \right)+{i\Pi }_{t}\left(\omega \right)$$

From Equations (11) and (12), we have
(22a)$$H\left(\omega \right)=\frac{\hbar }{2k_{B}}\Pi_{t}\left(\omega \right)$$
(22b)$$H_{\tau }\left(\omega \right)=\frac{\hbar }{2k_{B}}\Pi \left(\omega \right)$$

Hence,
(23a)$$H_{z}^{*}\left(\omega \right)\equiv H\left(\omega \right)-iH_{\tau }\left(\omega \right)=\frac{\hbar }{2k_{B}}\Pi_{t}\left(\omega \right)-i\frac{\hbar }{2k_{B}}\Pi \left(\omega \right)$$
(23b)$$H_{z}^{*}\left(\omega \right)=-i\frac{\hbar }{2k_{B}}\left(\Pi \left(\omega \right)+{i\Pi }_{t}\left(\omega \right)\right)=\frac{\hbar }{i}\frac{\Pi_{z}\left(\omega \right)}{2k_{B}}$$

Thus, we can see that a fundamental identity between the *complex-valued* Hamiltonian and the *complex-valued* entropy production is yielded by analytical continuation into the complex frequency plane $\hat{\omega }=-\left(\upsilon +i\omega \right)$ and, using the Cauchy–Riemann relations, the Hamiltonian and the entropy production are related by a Wick rotation and complex conjugation. Similarly, note that the Hamiltonian is usually associated with the reversible (imaginary time) *t*-axis, whereas the entropy production is usually associated with the irreversible (real time) *τ*-axis. In addition, we see that the complex entropy production (associated with thermodynamic irreversibility) is Wick-rotated (and complex-conjugated) with respect to the complex Hamiltonian (associated with thermodynamic reversibility). Thus, in a slightly more compact form, we can write the following:(23c)$$iH_{z}^{*}=\frac{h}{4\pi k_{B}}\Pi_{z}$$
where it is the (Wick-rotated) *complex conjugate* of the (complexified) “entropic Hamiltonian” of [4] that is reinterpreted here (in holographic natural units—that is, over the whole 4π sphere) as simply the (complexified) entropy production.

This symmetrical (conjugate) relationship between the Hamiltonian and entropy production underlines again the unity of the physical phenomena of thermodynamic reversibility and irreversibility when viewed from the perspective of complex time. Both processes (mutually being Hilbert transforms of each other) are now seen to exhibit fundamental (Noether) conservation properties based on equivalent variational calculus principles. Using such methods, we also expect to obtain new insights into the acausal (random) phenomena associated with entropy production such as radioactive decay and the apparent indeterminism of quantum mechanical measurement.

Equation (23c) also indicates that, although the complex time plane is defined by the (reversible) *t*-axis and the orthogonal (irreversible) *τ*-axis, these axes are not independent. That is to say, the Hamiltonian (usually defined empirically along the reversible *t*-axis) and the entropy production (usually defined empirically along the irreversible *τ*-axis) are essentially two sides of the same coin, being Hilbert-transform-related (see Equations (14)–(17)). What this means is that the choice of which axis to use to *fully* describe any physical phenomenon is arbitrary, depending only on the choice of metric. Either metric (+−−− or −+++, when considering the whole of 1 + 3 Minkowski spacetime) may be used, provided it is consistently applied. Conventionally (and here), the reversible metric for time, the first component in (−+++), is employed, indicating its imaginary nature, where the energy Hamiltonian is the physical quantity (with real measurable values) that is used to quantify the phenomenon, and the entropy production (emerging now as the conjugate physical quantity of interest) is treated as an imaginarily valued quantity (when viewed from the reversible *t*-axis). However, if the inverse metric (+−−−) is used for time, then the entropy production becomes the ‘real-valued’ physical quantity to be measured, with the energy Hamiltonian now imaginary. Both descriptions are equally valid, but our analysis shows that, once a metric is adopted, the primary associated temporal axis is thereby inevitably defined, and either Equation (14) or Equation (18) may be employed, but never both in the same analysis.

That is, we can now simplify the relevant (cross-axial) components of the complex Hamiltonian and entropy production in Equation (8):(24a)$$H_{z}=-\frac{\partial S_{cl}}{\partial t}+i\frac{\partial S_{cl}}{\partial \tau }\equiv H+i\frac{\hbar }{2k_{B}}\Pi$$
(24b)$$\Pi_{z}=\frac{\partial S_{th}}{\partial \tau }+i\frac{\partial S_{th}}{\partial t}\equiv \Pi +i\frac{2k_{B}}{\hbar }H$$

Similarly, we express the Hilbert transform relationships between the Hamiltonian and entropy production, simplifying Equations (13) and (17) and, for convenience (being more conventional and familiar), employing the imaginary (reversible) ω-component of the complex frequency:(25a)$$\Pi \left(\omega \right)=-\frac{2k_{B}}{\pi \hbar }P\overset{\infty }{\underset{-\infty }{\int }}\frac{H\left(\omega^{\prime }\right)}{\omega^{\prime }-\omega }d\omega^{\prime }$$
(25b)$$H\left(\omega \right)=\frac{\hbar }{2\pi k_{B}}P\overset{\infty }{\underset{-\infty }{\int }}\frac{\Pi \left(\omega^{\prime }\right)}{\omega^{\prime }-\omega }d\omega^{\prime }$$

Similar relations exist for the real (irreversible) ν-component of the complex frequency:(25c)$$\Pi \left(\nu \right)=-\frac{2k_{B}}{\pi \hbar }P\overset{\infty }{\underset{-\infty }{\int }}\frac{H\left(\nu^{\prime }\right)}{\nu^{\prime }-\nu }d\nu^{\prime }$$
(25d)$$H\left(\nu \right)=\frac{\hbar }{2\pi k_{B}}P\overset{\infty }{\underset{-\infty }{\int }}\frac{\Pi \left(\nu^{\prime }\right)}{\nu^{\prime }-\nu }d\nu^{\prime }$$

Thus, our analysis interprets the concept of imaginary energy as the imaginary component of the (complex) Hamiltonian, equivalent to (a real) entropy production; similarly, any imaginary component of the (complex) entropy production can simply be regarded as an energy term.

## 4. Application: The Alpha Particle

A system such as the alpha particle, which is unconditionally stable (it does not decay; the alpha has a QGT treatment [8]), comes into existence at some time in the past (at its creation) and then has a *constant* (non-time-varying) energy Hamiltonian $H_{\alpha }$. It is also independent of frequency, so its associated entropy production $\Pi_{\alpha }\;$must also be zero (as expected for an unconditionally stable system; see the Hilbert transform of Equation (25a)). Note that, being reversible, the Lagrangian line integral (in the *z*-plane) for the alpha remains firmly parallel to the temporal *t*-axis associated with reversibility. In the case of a freely moving alpha particle, its Hamiltonian (in the absence of any potential fields) is given below by Equation (26a), where *m* is the alpha particle mass and *p* is its momentum, such that its associated entropy production is given by Equation (26b) (using Equation (25a)):(26a)$$H_{\alpha }=\frac{p^{2}}{2m}=-\left(\frac{\hbar^{2}}{2m}\right)\nabla^{2}$$
(26b)$$\Pi_{\alpha }\left(\omega \right)=-\frac{2k_{B}}{\pi \hbar }P\overset{\infty }{\underset{-\infty }{\int }}\frac{H_{\alpha }}{\omega^{\prime }-\omega }d\omega^{\prime }=-\frac{2k_{B}H_{\alpha }}{\pi \hbar }P\overset{\infty }{\underset{-\infty }{\int }}\frac{1}{\omega^{\prime }-\omega }d\omega^{\prime }=0$$

This is because the Hilbert transform of a constant quantity is simply zero (according to the unconditional stability of the alpha particle). The inverse Hilbert transform can also be undertaken to yield the alpha Hamiltonian. However, the inverse Hamiltonian of zero is also zero. In this case, however, similar to the case of the real part of the refractive index, the Hilbert transform (Kramers–Kronig) relations only refer to the *variable* part of the overall physical expression and ignore any constant aspects of the physical quantity. In this case, the more general expressions for the Hamiltonian and entropy production of a process should also include constant (d.c.) components, thus
(27a)$$\Pi \left(\omega \right)=\Pi_{0}-\frac{2k_{B}}{\pi \hbar }P\overset{\infty }{\underset{-\infty }{\int }}\frac{H\left(\omega^{\prime }\right)}{\omega^{\prime }-\omega }d\omega^{\prime }$$
(27b)$$H\left(\omega \right)=H_{0}+\frac{\hbar }{2\pi k_{B}}P\overset{\infty }{\underset{-\infty }{\int }}\frac{\Pi \left(\omega^{\prime }\right)}{\omega^{\prime }-\omega }d\omega^{\prime }$$
where the quantities $\Pi_{0}$ and *H*_0_ (with the subscript ‘0′ signifying the zero-frequency or d.c. value) are constants independent of any frequency variation. For the (unconditionally stable) alpha particle, we must also have $\Pi_{0}$ = 0. In this case, the inverse Hilbert transform of $\Pi_{\alpha }\left(\omega \right)$ (the second term associated with the RHS of Equation (27b)) is zero, so the Hamiltonian of the alpha is simply given by the constant *H*_0_ (independent of frequency): *H*_0_ ≡ *H*_α_ = *p*^2^/2*m* = −(ℏ^2^/2*m*)∇^2^ (see Equation (26a)).

## 5. Application: A Decaying Harmonic Oscillator

Here, we consider the simple case of a decaying oscillator. It has been noted that, even though this is a system with friction, it can be described with a Hamiltonian (and Lagrangian) with the implication that Liouville’s theorem applies. It is frequently (wrongly) assumed that a dissipative system cannot be successfully described with a Hamiltonian (Lagrangian) within the context of Liouville’s theorem. However, from a kinematic perspective, this is clearly not the case, as already elegantly recognised by Kirk McDonald [18], whose treatment we follow here. Indeed, this is also already described from the entropic (QGT) perspective, where dissipative systems (with a finite entropy production) have already been successfully described from the entropic (QGT) perspective using an *entropic* Hamiltonian and Lagrangian within the context of the *entropic* Liouville theorem [7].

This current paper, where the entropy production is found to be the imaginary counterpart to the (real) kinematic component of the Hamiltonian (with both quantities related to each other by the Hilbert transform), carries the implication that, rather than being subject to two independent applications of Liouville’s theorem (each with associated Hamiltonians and Lagrangians), the two frameworks are actually two sides of the same coin; that is, together they obey a unified Liouville’s theorem that covers both the real and imaginary (reversible and irreversible) aspects of a single Hamiltonian (and Lagrangian) description of any system.

Using McDonald’s analysis (but defining the damping constant, in advance, as representing a dissipative term, so that the negative sign is explicitly indicated), we consider a damped harmonic oscillator (mass on a spring) with the equation of motion:(28)$$\ddot{x}-\alpha \dot{x}+\omega_{0}^{2}x=0$$
where *x* is the spatial coordinate and the parameter α=β/m, where β is the damping constant, *m* is the mass, and *k* is the spring constant, with $\omega_{0}^{2}=k/m$ representing the natural frequency of the system. McDonald demonstrates that the Hamiltonian can be represented by the following:(29)$$H=\left(T+V\right)e^{-\alpha t}=\frac{m{\dot{x}}^{2}+kx^{2}}{2}e^{-\alpha t}=Ue^{-\alpha t}$$
where *U* = *T* + *V* is the conventional Hamiltonian (the sum of the kinetic *T* and potential *V* energies) for a dissipationless system. Having defined a Hamiltonian for the dissipative system, we can now use it to derive the associated entropy production and demonstrate consistency with the key thrust of this paper.

We first define the system to be causal, that is, to have a specific point in time when it comes into existence. For simplicity, we define *H*(*t*) = 0 for *t* < 0. In this case, we calculate the Fourier transform of *H*(*t*) to determine its real and imaginary frequency components:(30)$$H_{z}\left(\omega \right)=\int_{-\infty }^{\infty }H\left(t\right)e^{i\omega t}dt=U\int_{0}^{\infty }e^{-\alpha t}e^{i\omega t}dt=\frac{U}{\alpha -i\omega }$$

It is clear that the frequency components of the Hamiltonian have real and imaginary components:(31a)$$H_{r}\left(\omega \right)=\frac{\alpha }{\alpha^{2}+\omega^{2}}\;U$$
(31b)$$H_{i}\left(\omega \right)=\frac{\omega }{\alpha^{2}+\omega^{2}}U$$

It is immediately apparent that the two components of Equation (31) are Hilbert transforms of each other (as required by causality). Relating them now directly to the (real) Hamiltonian H(ω) and (real) entropy production Π(ω), defined within the main part of the paper (Equations (10), (13), and (14)), we now explicitly write the following:(32a)$$H\left(\omega \right)=\frac{\alpha^{2}}{\alpha^{2}+\omega^{2}}\;U\;\;\;$$
(32b)$$\Pi \left(\omega \right)=\frac{2k_{B}}{\hbar }\frac{\omega \alpha }{\alpha^{2}+\omega^{2}}U$$
where we have included an additional instance of the frequency α (common to both expressions) in order to maintain correct dimensionality.

From Equation (24a), the complex Hamiltonian associated with a decaying oscillator is thus as follows:(33)$$H_{z}\left(\omega \right)=H\left(\omega \right)+i\frac{\hbar }{2k_{B}}\Pi \left(\omega \right)=\frac{\alpha U}{\alpha^{2}+\omega^{2}}\left(\alpha +i\omega \right)$$

More specifically, we can also integrate the entropy production (Equation (32b)) across the frequency domain and calculate the average rate of change in entropy associated with the damped oscillator. The characteristic time *τ*_0_ for the exponential decaying system is given by *τ*_0_ = 1/α, so that the appropriate ‘cut-off’ frequency for the integral (below which we can assume most of the entropy production is associated) is α, such that we simply divide the integral by α (as indicated below) to determine the average entropy production. According to the usual conventions of measurement theory, we only consider the positive frequency components:(34a)$$\bar{\Pi }=\frac{1}{\alpha }\int_{0}^{\infty }\frac{2k_{B}}{\hbar }\frac{\omega \alpha }{\alpha^{2}+\omega^{2}}Ud\omega =\frac{2k_{B}}{\hbar }U{\left[\frac{1}{2}\ln\left(1+{\left(\frac{\omega }{\alpha }\right)}^{2}\right)\right]}_{0}^{\infty }$$

Again, exploiting the appropriate ‘cut-off’ frequency α means that Equation (34a) can be closely approximated:(34b)$$\bar{\Pi }\approx \frac{2k_{B}}{\hbar }U{\left[\frac{1}{2}\ln\left(1+{\left(\frac{\omega }{\alpha }\right)}^{2}\right)\right]}_{0}^{\alpha }=\frac{k_{B}}{\hbar }U\ln2$$

The total entropy *S* produced by the decaying oscillator over the exponential characteristic time $\tau_{0}$ is thus as follows:(35)$$S=\bar{\Pi }\tau_{0}=\frac{k_{B}}{\alpha \hbar }U\ln2$$

We can assume all the energy of the decaying oscillator is dissipated over the characteristic time $\tau_{0}$ (given the usual conventions for treatments of exponentially decaying systems), and then the associated temperature T is straightforwardly given by the following:(36)$$T=\frac{\partial U}{\partial S}=\frac{\alpha \hbar }{k_{B}\ln2}$$

It is interesting to note from Landauer’s principle [19] that the minimum quantity *k_B_*Tln2 of energy is associated with the erasure (or loss) of a bit of information, where a bit is also equivalent to a system degree of freedom (see also [8]). Thus, it is clear that, for a decaying oscillator, as its energy is dissipated, Equation (36) suggests that each of its degrees of freedom dissipates a quantity of energy αℏ. Clearly, when the oscillator is dissipationless and α = 0, then, also according to Equation (36), none of its degrees of freedom are dissipating any energy.

## 6. Application: The Black Hole

The opposite system to the alpha particle (featuring an unconditionally *irreversible* physical phenomenon with a constant entropy production) must also have a Hamiltonian with zero variation with frequency. Such a physical system is represented by a black hole of mass *M*_BH_, which has a constant entropy production associated with the Hawking radiation (HR, see Parker and Jeynes [5] Equation (25)):(37)$$\Pi_{HR}=\frac{c^{3}k_{B}}{2GM_{\mathrm{BH}}}$$
where *G* is the gravitational constant and *c* is the speed of light. Of course, the Hawking radiation has a frequency distribution (spectral radiance), obeying the black body radiation law:(38)$$B\left(\omega \right)=\frac{\omega^{2}}{2\pi^{2}c^{2}}\frac{\hbar \omega }{e^{\hbar \omega /k_{B}T}-1}$$
which describes the energy spectrum (in effect, the Hamiltonian). Unfortunately, calculating the Hilbert Transform of Equation (38) to obtain the associated entropy production spectrum is not analytically easy. However, Parseval’s Theorem allows us to calculate the Hamiltonian associated with the Hawking radiation because, being Hilbert transforms of each other, we can use this theorem to equate the two quantities associated with the Hamiltonian and the entropy production:(39)$$\int_{-\infty }^{\infty }{\left|H\left(\omega \right)\right|}^{2}d\omega =\frac{\hbar^{2}}{4k_{B}^{2}}\int_{-\infty }^{\infty }{\left|\Pi \left(\omega \right)\right|}^{2}d\omega$$

Assuming that the entropy production of Equation (37) is appropriately proportional to the integrated spectrum entropy production, where Ω represents an equivalent spectral width, we can write the following:(40)$$\Pi_{HR}^{2}\Omega \equiv \int_{-\infty }^{\infty }{\left|\Pi \left(\omega \right)\right|}^{2}\left(\omega \right)d\omega$$

Then, it is clear the Hamiltonian associated with the integrated Hawking radiation can be given by the following:(41)$$H_{HR}^{2}\Omega \equiv \int_{-\infty }^{\infty }{\left|H\left(\omega \right)\right|}^{2}d\omega =\frac{\hbar^{2}}{4k_{B}^{2}}\Pi_{HR}^{2}\Omega ={\left(\frac{\hbar \Pi_{HR}}{2k_{B}}\right)}^{2}\Omega$$

In this case, the integrated energy of the Hawking radiation radiated away by the BH is given by the following:(42)$$H_{HR}=\frac{\hbar \Pi_{HR}}{2k_{B}}=\frac{\hbar c^{3}}{4GM_{\mathrm{BH}}}$$

Substituting Equation (37) into Equation (27b) yields the following:(43)$$H\left(\omega \right)=H_{0}+\frac{\hbar }{2\pi k_{B}}P\overset{\infty }{\underset{-\infty }{\int }}\frac{\Pi }{\omega^{\prime }-\omega }d\omega^{\prime }=H_{0}+\frac{\hbar \Pi_{0}}{2\pi k_{B}}P\overset{\infty }{\underset{-\infty }{\int }}\frac{1}{\omega^{\prime }-\omega }d\omega^{\prime }=H_{0}$$

The Hamiltonian for a stationary black hole is conventionally given by its mass *M*_BH_ within the Schwarzschild radius; yet, as is well known, any Hamiltonian can also be offset by a constant quantity (with there being no absolute value for the energy, see Caticha 2021 [20]), so that the value of the associated Hamiltonian as determined by Equation (27b) can be additionally offset by the energy lost to the Hawking radiation *H*_0_ = *M*_BH_*c*^2^ − *H_HR_* as required. Note that such an offset (by unity) is also seen in the Kramers–Kronig expression for the real part of the refractive index to obtain the correct Hilbert transform relationships (see, as an example, the unity offset in Equation (1.1) of Toll [16]). That is to say, we can now rewrite Equation (43) as follows:(44)$$H=M_{\mathrm{BH}}c^{2}-\frac{\hbar c^{3}}{4GM_{\mathrm{BH}}}+\frac{\hbar \Pi_{0}}{2\pi k_{B}}P\overset{\infty }{\underset{-\infty }{\int }}\frac{1}{\omega^{\prime }-\omega }d\omega^{\prime }=M_{\mathrm{BH}}c^{2}-\frac{\hbar c^{3}}{4GM_{\mathrm{BH}}}$$

Of course, the energy lost to the BH by the Hawking radiation is insignificant compared with the energy due to the BH mass, so it is generally safe to neglect the *H_HR_* term on the RHS of Equation (44).

Intrinsic to our complex time analysis is the possible existence of two distinct values each for *H*_z_ and $\Pi_{z}$, which are normally degenerate, but which may become apparent at the extreme physical conditions of a black hole. Thus, Equations (22) suggest the following identities for the ‘trans-axial’ quantities, which may become apparent at the black hole event horizon where the metric is assumed to invert:(45a)$$\Pi_{t}=\frac{2k_{B}}{\hbar }H\;$$
(45b)$$H_{\tau }=\frac{\hbar }{2k_{B}}\Pi$$

Substituting the black hole Hamiltonian Equation (44) into Equation (45a) provides us with an entropy production $\Pi_{t}$ associated with the black hole that is considerably larger than that associated with the Hawking radiation:(46)$$\Pi_{t}=\frac{2k_{B}}{\hbar }M_{\mathrm{BH}}c^{2}$$
where we have also chosen to neglect in Equation (46) the *H_HR_* Hawking radiation term in Equation (44). Actually, substituting the Hawking radiation Equation (37) into Equation (45b) now directly offers us the associated (integrated) energy Hamiltonian of the (very small) Hawking radiation:(47)$$H_{\tau }=\frac{\hbar }{2k_{B}}\frac{c^{3}k_{B}}{2GM_{\mathrm{BH}}}=\frac{\hbar c^{3}}{4GM_{\mathrm{BH}}}\left(\equiv H_{HR}\right)$$

Indeed (as a cross-axial Hamiltonian), this is why in Equation (44) it appears to have a negative sign, because it is energy that is being lost to the system over time. Associated with the extreme physics of black holes (especially the supermassive BHs at galactic centres) are the so-called Planck quantities, and it is thus no surprise that we see both the Planck energy *E*_P_ and what we could call the “Planck entropy production” $\Pi_{P}$ intimately connecting the conventional and trans-axial Hamiltonian and entropy production quantities, respectively:(48)$$H.H_{\tau }=\frac{\hbar c^{5}}{4G}=\frac{E_{P}^{2}}{4}$$
(49)$$\Pi .\Pi_{t}=\frac{c^{3}k_{B}}{2GM_{\mathrm{BH}}}.\frac{2k_{B}}{\hbar }M_{\mathrm{BH}}c^{2}=\frac{c^{5}}{\hbar G}k_{B}^{2}={\left(\frac{k_{B}}{t_{P}}\right)}^{2}\equiv \Pi_{P}^{2}$$
where *t*_P_ is the Planck time. The Planck entropy production quantity $\Pi_{P}$ is the same as the entropy production term given previously (Parker and Jeynes [5] Equation (31); the factor 2π comes from an ambiguity in the definition of wavenumber) in the context of the entropy production of spiral galaxies. Thus, we find that, associated with a black hole, there exists another very large entropy production term $\Pi_{P}$ (Equation (49)) that is 46 orders of magnitude larger than the term associated with the Hawking radiation $\Pi_{HR}$ (Equation (37)); using the example of the Milky Way featuring a supermassive BH of mass 4.3 × 10^6^ solar masses), and which can be understood as being related to the highly energetic processes seen occurring in the accretion disk surrounding a black hole.

## 7. Discussion

Previously, entropy production has been simply identified with the Hamiltonian (in natural units; see Equation (15) in [4]). In this complexified analysis, we now identify the entropy production with the (Wick-rotated) complex conjugate of the Hamiltonian (Equation (23c)). The purpose of the present analysis is to make full use of the power of the Hilbert transform, which requires a fully complexified formalism.

We note that, in order for the entropy production to be simply the Wick-rotated complex conjugate of the Hamiltonian (in natural units; Equation (23c)), the Planck constant must be reduced by 4*π* (rather than the conventional 2*π*). On the one hand, reducing by 4*π* is appropriately suggestive of the holographic basis of the QGT analysis (and would also avoid the apparent ‘paradox’ of half-integer spins in a quantum theory); however, rejigging quantum theory to use a 4*π*-reduced Planck constant in order to achieve consistent natural units between the kinematic and entropic domains would cause confusion in the conventional (historical and widely accepted) formalisms of kinematic quantum theory. Rather, the factor of two must thus be simply understood and accepted as a historical fact from the development of quantum theory, but which does not undermine the assertion of the natural relationship between the entropy production and the conventional Hamiltonian. The alpha particle and the black hole each represent unitary physical systems (both requiring only four parameters for their full description), yet the former is unconditionally reversible, while the latter is unconditionally irreversible. It is interesting to note that such complementary physical phenomena both lend themselves to the Hilbert transform analysis of the expressions for their Hamiltonian and entropy production (provided, of course, these are complexified).

The well-known Kramers–Kronig dispersion relations are directly derived from the Hilbert transform pairs, but based on additional symmetry constraints (the real function being even for real frequencies, whereas the imaginary function is odd; the so-called *crossing conditions* in quantum mechanical scattering). With a minimum of assumptions about the physical process under investigation, the Kramers–Kronig relations provide a complete description between two conjugate phenomena using only the empirically accessible positive frequencies, and their application to the system Hamiltonian and entropy production is thus the subject of future work.

The metric of complex time also has interesting implications in the context of black hole thermodynamics. In particular, it is clear that ‘conventional’ reversible (imaginary) time has a metric of a negative sign (−+++), whereas the irreversible (real) time is associated with a metric of a positive sign (+−−−). Normally, the sign of the metric is indicative of whether the dimension is timelike (negative) or spacelike (positive) in character. However, here, we see that the sign of the metric is also indicative of whether a physical phenomenon is thermodynamically reversible or irreversible. It is well known that the metric changes its sign either side of the vicinity of the event horizon of a black hole. Misner et al. [21], in interpreting the sign change, pose the following question: ‘*What does it mean for r [radius] to “change in character from a spacelike coordinate to a timelike one”?’* and offer the most obvious answer as “*the reversal there of the roles of t [time] and r as timelike and spacelike coordinates.*” However, we now see an alternative interpretation as the metric changes sign; that is, the spacetime geometry changes from an intrinsically reversible geometry to a thermodynamically irreversible geometry. Interestingly, this is still consistent with the conventional interpretation of the unforgiving ‘point of no return’ beyond the event horizon. However, now we may couch it in specifically thermodynamic terms, highlighting the intrinsic irreversibility of the geometry beyond the event horizon (with the inevitable increase in entropy according to the Second Law). This has the interesting corollary that the role reversal of the spacelike and timelike coordinates may no longer be strictly necessary. However, it is clear that inverting the metric across the event horizon can now be simply interpreted as the transformation of what were reversible processes into irreversible processes (and *vice versa*), with the additional interesting implication that what were irreversible phenomena this side of the event horizon become reversible ones beyond it. The implications of such a *thermodynamic inversion* at the event horizon lie beyond the scope of this work.

Phenomenologically, the conventional Hamiltonian of a system (representing the energy) is assumed to be positive-definite (like the inertial mass) because it is considered from the perspective of the real (reversible) time axis. Similarly, the entropy production (the rate of increase in entropy) considered from the perspective of the imaginary (irreversible) time axis is also positive-definite (as required by the Second Law). Conversely, considered from the reversible time axis, the entropy production is imaginary and, considered from the irreversible time axis, the energy is imaginary. Switching between the two axes defining the complex temporal plane is effected by a Wick rotation, as represented by the factor i in Equation (23c). To obtain the correct relationship between the system’s (complexified) Hamiltonian and its (complexified) entropy production (two equivalent descriptions, in natural units), a complex conjugation is also needed. In effect, Equation (23c) expresses an intrinsic redundancy; that is, either the (complexified) Hamiltonian or the (complexified) entropy production can be used to fully describe the evolution of the system, and each is sufficient by itself to completely define its evolution in time.

Note that, if purely real definitions of the Hamiltonian and entropy production are used, then *both* are required to fully describe a system. However, by allowing them to be complex-valued, all of the information of the ‘real’ entropy production becomes embedded into the imaginary component of the Hamiltonian; *vice versa*, all of the information of the ‘real’ Hamiltonian is to be found in the imaginary component of the entropy production. Together, this now allows us to interpret ‘imaginary energy’ as simply the entropy production associated with a process, and conversely, the ‘imaginary entropy production’ of a process is simply its energy content. The Hamiltonian (energy) of a system is usually defined as the total energy of an essentially lossless system (that is, all energy is explicitly accounted for and conserved according to the First Law), such that the energy is calculated to be purely real in nature. For such lossless (and thus reversible) systems, the entropy stays constant and the entropy production is zero. Allowing energy to be lost from a system due to dissipative processes requires a complex Hamiltonian with an associated finite entropy production.

We are so familiar with the (generally implicit) assumption of reversibility as the underlying description of a process that we automatically employ the ‘reversible’ (and, incidentally, imaginary, although this is generally neither explicitly acknowledged nor even recognised) temporal axis to describe physical processes (together with real energies); perhaps this is why it has taken so long to identify the “imaginary” counterpart of energy, the entropy production (also conserved). However, this is just convention. Had the physicists of the 19th century been able to explicitly quantify the dissipations of all parts of a process (rather than merely reasoning qualitatively that dissipation was simply an undesired result of “*not fulfilling the criterion of a ‘perfect thermodynamic engine’*” [22]), they might have identified the law of conservation of entropy production as the First Law of the Thermodynamics first, and then puzzled about the physical meaning of an imaginary aspect of the entropy production. We underline here the importance of the Minkowski space metric (−+++) and its implicit assumption of thermodynamic reversibility. The alternate metric (+−−−) has the implicit assumption of thermodynamic irreversibility. Note that Roger Penrose [23] (2004, §17.8) also points out that the choice of metric depends on one’s purpose.

It is clear that the physical time associated with any physical phenomenon can only be uni-valued as it evolves over time (monotonically increasing) in its trajectory across the complex temporal plane. That is to say, even though, here, we invoke the concept of a complex temporal plane, the real and imaginary components are not independent of each other, but must also obey the familiar Hilbert transform relations described here in detail. Any real time *τ* can only ever be identified with a single imaginary time *t*. Complex time (real or imaginary depending on the reversibility/irreversibility of the process) must increase inexorably according to the Second Law. However, in addition to describing time as complex, considerations of chirality cannot be neglected because trajectories (not in a straight line) across the temporal plane must locally tend to have either a clockwise or anticlockwise character (however weak), according to their local gradient or indeed inflection. A natural chirality to physical geometry has already been noticed from a QGT perspective, as Parker and Walker already showed in 2010 [24] that natural DNA must be right-handed (see also the more rigorous treatment of Parker and Jeynes [6] Appendix A) according to the Second Law. It is becoming clear that the complex conjugate operation can be thought to build chirality into the formalism (see, for example, the “Berry phase” [25] analysis by Laha et al. [26] of chiral waveguide responses). How such chirality in the complex time plane physically manifests itself (and allows itself to be observed) must continue to be a topic of further investigation.

Finally, in applying our approach to the alpha particle (associated with physics on the microscopic scale, see also [8]) and to a black hole (a macroscopic phenomenon, see also [5]) shows how QGT can successfully handle physical phenomena at all scales. In previous work [6], we have already indicated how QGT’s underlying basis in hyperbolic spacetime implicitly allows phenomena over all scales to be appropriately treated, although the hyperbolic (essentially, logarithmic) nature of entropic processes has not needed to be explicitly invoked in this paper. However, we see here that the physics of reversible and irreversible processes can now be elegantly integrated via the use of complex time into the microscopic scale, yet without a noticeable micro/macro boundary, such that macroscopic scale processes are also seamlessly handled. Together, this continues to indicate the unity of description for physical phenomena that geometrical thermodynamics (QGT) has to offer, particularly with regard to a theory of scale relativity [27].

[Note added in proof:] It is worth mentioning that Michael Berry [28] underlines both the value of the Wick rotation for “transforming” a diffusion problem into a problem of “quantum mechanics” (see our Equations (1) *passim*) and also the qualitatively different sorts of solutions that emerge in the reversible/irreversible cases. Indeed he points out the futility in expecting to be able to intuitively anticipate the resulting physical phenomena from such a Wick rotation: “…*the intuition is wrong, dead wrong*”. In the irreversible case he describes (relating to the sex lives of moths) how the analytical solutions involve completely different sorts of singularities from those obtained in the reversible cases (as in optics examples, although these also may be deeply intricate due to additional issues of irreversibility—see the context of Figure 2 of reference [25]).

## 8. Conclusions

In a fully complexified treatment, we have shown essentially that the (complex) Hamiltonian is the Wick-rotated complex conjugate of the (complex) entropy production (in natural units). We have considered the application of this treatment to three canonical cases: the alpha particle [8] and the black hole [5] (both unitary entities in QGT), as well as the real (dissipative) harmonic oscillator [18]. Landauer’s principle is recovered by an explicit treatment of the latter case, which indicates a minimal amount of irreversibility to be associated with any dissipative system whose Hamiltonian and entropy production (by virtue of causality) are intrinsically related to each other via the Hilbert transform. In QGT, the alpha and the black hole (BH) have closely related treatments, because, as a consequence of the entropic Liouville Theorem [7], the Bekenstein–Hawking equation applies equally to both. We demonstrate that the alpha (trivially reversible with zero entropy production) is treated coherently. The BH is the opposite case (unconditionally irreversible) and we demonstrate that the previous entropy production results are also obtained using these methods.

Just as energy and entropy are closely related, so the Hamiltonian and the entropy production are also closely related. However, whereas the conventional thermodynamics considers energy and entropy to be the most-closely related (and thereby quasi-isomorphic) quantities, simply related to each other via the temperature acting as a coupling coefficient, in this work, we find that it is the entropy production (rate of increase in entropy) that is the actual isomorph to energy.

Using analytical continuation into a fully complexified representation (in which time also is complex), we show that the natural (Hilbert transform) expressions of the various quantities display these close relations very clearly. This has allowed a detailed calculation of the entropy production of black holes (consistent with a previous such calculation [5], but apparently independent of it), as a simple example of a system with a non-zero entropy production.

Fully complexifying the formalism with appropriate consideration of causal properties immediately allows the application of the Hilbert transform, which in turn enables very simple relationships to be displayed. In particular, irreversible systems are treated in exactly the same way as reversible ones; this allows irreversible systems to be treated as such (invoking their own conservation law based on the application of Noether’s theorem to quantities related to the entropy production) rather than having to use perturbation theory on the formalisms for reversible systems. Moreover, of course, it is well known that application of perturbation theory, on its own, can never recapture the full richness of mathematical behaviours associated with analytic functions, which feature local and global (essentially, non-local) characteristics (holomorphism) and allow the process of analytic continuation.

Understanding the intrinsic role of thermodynamic irreversibility along with its conservation law in the physics of the universe, over and above the conventional understandings relating to energy and its conservation, will help provide critical insights into the resolution of at least some of the paradoxes currently recognized within contemporary physics.

## Data Availability

This work is analytical: no data were used or generated.

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
