# Peer review of "Relating a System’s Hamiltonian to Its Entropy Production Using a Complex Time Approach"

_entropy, 2023, doi:10.3390/e25040629_

Round 1
Reviewer 1 Report
This work deals with, perhaps, the most fundamental, most controversial and still unresolved issue in modern science --- the arrow of time. It capitalizes on similarity of statistical expression for entropy and quantum action (it would be good to mention the path integral formulation of quantum mechanics in this context), which indeed may have profound physical implications. This similarity is revealed by introduction of complex time, which is conceptually connected to reversible and irreversible processes in physics.
While I believe that pioneering works should be published even if they do not offer a complete picture, I have some specific remarks that need to be addressed.
- There is a bit of logical gap between equations (1-2) and (4)
- The idea that irreversibility is intrinsic for matter has been around for quite a some time (see SN Applied Sciences, 3 (7) 710)
- It is essential to provide at least one illustration where the suggested complex methodology works and gives a reasonable prediction. Assuming that real and imaginary times have the same scaling, the suggested link between Hamiltonian and entropy production implies some minimal amount of irreversibility is to be attached to interacting systems. Can it be estimated and compared to reality?
- Alpha-particle case represents trivial zero solution for entropy production and therefor is not characteristic for the theory. There need to be some interactions to reveal irreversibility.
- While black hole seems to be a good case, the link between energy (mass) and entropy production seems not to work particularly well for this case and additional energy and additional production are introduced. The role of these additional variables is not easy to understand. Maybe black holes are not the best case to illustrate this theory.
Reviewer 2 Report
This paper explores the idea that time is better represented by a complex plane than a real variable. Using this, the authors derive relations between the Hamiltonian of a system and its rate of entropy production. This general treatment, set out in sections 1 to3, seems to be correct, but it is not clear how it adds to our understanding of actual physical processes, although the passing reference to quantum measurement is intriguing.
Section 4 discusses the application of their ideas to the case of an alpha particle. A this is a stable system the entropy production must be zero and this is indeed what they find. It is not clear why they specifically use the alpha particle as their discussion applies equally to any stable system. This should be made clear at the beginning of this section, rather than later in the paper.
I have some problems with section 5 which applies the analysis to the properties of a black hole. The authors first show that the BH Hamiltonian is time independent and equal to Mc2 - which would seem to put it on a par with other stable systems, such as the alpha particle discussed in section 4. In the presence of Hawking radiation the BH is steadily losing mas as well as producing entropy, so the Hamiltonian must be time dependent, but this has not been taken into account. Later in this section, they derive an alternative expression for the H and they use the product of the two expressions to represent H‑2 in (29), which implies that they must be equal, although this is not established in the paper. They then go on to derive an alternative expression for the rate of entropy production that is 46 orders of magnitude greater than that associated with Hawking radiation. They attribute this to “the highly energetic processes seen occurring in the accretion disk surrounding a black hole”. But, if the BH is in the process of accreting, this must make a positive contribution to the rate of mass change, which is not considered. If they mean that this entropy production is from the accretion disk, then it is clearly not from the black hole itself. The paper states that the same expression for the entropy production was obtained in ref. 4 eq. 31. There is no reference 31 in this paper, which relates to the structure of galaxies.
The paper would be clearer if the usual symbol, k B, were used for the Boltzmann constant.
Round 2
Reviewer 1 Report
The authors responded to my questions and introduced an additional example. I recommend publication.
Reviewer 2 Report
The authors have gone some way to address my earlier criticisms. However, I still have problems with the section on the black hole. I should make it clear that I do not have specialist expertise in this area and the following remarks are based on my general physics knowledge and the apparent inconsistency between this section and earlier parts of the paper. If the theory developed in the paper is correct, I would expect that the Hamiltonian obtained from (39) should include a term associated with the mass loss due to Hawking radiation which has a black body spectrum, and that this should equal the Hilbert transformation of the entropy production. If this is ignored, then the expression for the entropy production obtained from the Hamiltonian should be zero – just as it is for the alpha particle in (26a) – while if it is included, I should expect it to produce a term proportional the Hilbert transform of the black-body spectrum. Instead, the authors derive an expression for the entropy production that is 46 orders of magnitude greater then they give in (38). They speculate that this may be associated with the properties of an accretion disk. In their reply to my earlier report, they state that “without a black hole there is no accretion disc”. I think they mean the opposite: i.e., “without an accretion disc there is no black hole”, but even if this is true, the existence of the accretion disk has not been included in their initial description of the black hole so their is no reason why it should emerge here. The authors should revise this section in the light of the above and acknowledge that its conclusions may indicate a fundamental inconsistency in their approach.
Reviewer 3 Report
see comments

Round 3
Reviewer 2 Report
The authors have addressed the points I raised previously, so I can now recommend acceptance.